

# Expression patterns and the prognostic value of the EMILIN/Multimerin family members in low-grade glioma

Yonghui Zhao, Xiang Zhang, Junchao Yao, Zhibin Jin and Chen Liu

Department of Neurosurgery, Cangzhou Central Hospital, Cangzhou, Hebei Province, China

## ABSTRACT

Managing low-grade gliomas (LGG) remains a major medical challenge due to the infiltrating nature of the tumor and failure of surgical resection to eliminate the disease. EMILIN/Multimerins contain the gC1q signature, which is involved in many tumor processes. However, the expression and prognostic value of EMILIN/Multimerins in LGG remains unclear. This study used integrated bioinformatics analysis to investigate the expression pattern, prognostic value and function of EMILIN/Multimerins in patients with LGG. We analyzed the transcription levels and prognostic value EMILIN/Multimerins in LGG using the ONCOMINE, Gene Expression Profiling Interactive Analysis (GEPIA) and UALCAN databases. The mutation and co-expression rates of neighboring genes in EMILIN/Multimerins were studied using cBioPortal. TIMER and Metascape were used to reveal the potential function of EMILIN/Multimerins in LGG. According to our analysis, most EMILIN/Multimerins were overexpressed in LGG and shared a clear association with immune cells. GEPIA analysis confirmed that high levels of EMILIN/Multimerins, not including MMRN2, were associated with a poor prognosis in disease-free survival of patients with LGG. Additionally, we discovered that EMILIN/Multimerins may regulate LGG and we found a correlation between their expression patterns and distinct pathological grades. We found that EMILIN/Multimerins serve as possible prognostic biomarkers and high-priority therapeutic targets patients with LGG.

# INTRODUCTION

Low-grade gliomas (LGG), comprised of WHO grades II and III gliomas (*Louis et al., 2007*), account for 15–20% of all central nervous system glial tumors (*Forst et al., 2014*). Previous studies show LGG is highly heterogeneous in pathology, molecular features and prognosis (*Burak Atci, Yilmaz & Samanci, 2019*). Although LGG have a benign course with longer-term survival in comparison with glioblastoma (*Brat et al., 2015*), LGG cannot be cured entirely by conventional treatment, due to the frequent cancer recurrence and malignant progression which can turn into high-grade glioma (*Kumthekar, Raizer & Singh, 2015*). It is imperative for early LGG detection and precise prognosis that we screen for sensitive and reliable biomarkers.

Corresponding author
Yonghui Zhao,
zhaoyonghui_1987@126.com

The recently defined elastin microfibrillar interface protein (EMILIN) and Multimerin family is characterized by a C-terminal gC1q globular domain, containing five members: EMILIN1, EMILIN2, EMILIN3, MMRN1 and MMRN2 (*Colombatti et al., 2012*; *Schiavinato et al., 2012*). These members have exerted a diverse range of function in the previous study (*Capuano et al., 2019*; *Rabajdova et al., 2016*; *Schiavinato et al., 2016*), which can affect cell growth, wound healing, angiogenesis, lymphangiogenesis and tumor microenvironment, etc. Notably, previous research has reported that high expression of EMILIN3 was confirmed to predict poor survival in LGG (*Zeng et al., 2018*). Also, recent studies have confirmed that the appearance of Multimerin-2 and EMILIN-2 is significantly altered in gastric cancer patients (*Andreuzzi et al., 2018*) and EMILIN2 has a bidirectional role in tumor micro environments (*Mongiat et al., 2010*). Although EMILIN/Multimerins act as promising biomarkers in tumor development, heir expression pattern, molecular function and prognostic value in LGG have not been investigated.

We used bioinformatics analysis to investigate the expression pattern, molecular mechanism and prognostic value of EMILIN/Multimerins in patients with LGG. The expression of most EMILIN/Multimerins was significantly upregulated in patients with LGG. We found that a low expression of EMILIN/Multimerins predicted higher survival rates, establishing prognostic value. Additionally, we studied the potential for EMILIN/Multimerins to enhance molecular diagnosis and predict LGG prognosis.

## METHODS

### Oncomine analysis

Oncomine analysis (*Rhodes et al., 2004*) was used to explore EMILIN/Multimerin expression levels in various cancer types. In the first search, we used the following keywords: "EMILIN1", "EMILIN2", "EMILIN3", "MMRN1" and "MMRN2", the primary filter for the analysis type was "Cancer vs. Normal Analysis", and the chosen cancer type was "Brain and CNS Cancers". We set the following thresholds: $p$-value = 0.05, fold change "All" and gene rank Top 10%. To compare the expression of EMILIN/ Multimerins in LGG subtypes and normal tissue, "EMILIN1" and "EMILIN2" were selected as keywords. "Diffuse Astrocytoma vs. Normal Analysis" was the chosen analysis type, and we set thresholds for $p$-value = 0.05, fold change "2" and gene rank Top 10%. Additionally, "EMILIN3", "MMRN1" and "MMRN2" were selected as keywords, and the analysis type and threshold were the same as the previous search (Diffuse Astrocytoma vs. Normal Analysis; $p$-value = 0.05, fold change "2" and gene rank Top 10%).

### UALCAN analysis

Using the UALCAN database (*Chandrashekar et al., 2017*), we analyzed the expression profiles of EMILIN/Multimerin expression patterns in normal brain tissue and low-grade glioma samples. We established the correlation between expression levels of candidate genes and LGG tumor grades. The unique transcription expression of each tumor grade was compared using student's $t$-test, considering $p < 0.01$ statically significant.

### Gene expression profiling interactive analysis

We conducted Gene Expression Profiling Interactive Analysis (GEPIA) using The Cancer Genome Atlas (TCGA) database (*Tang et al., 2017*) to investigate EMILIN/Multimerin expression levels on overall survival (OS) and disease-free survival (DFS) in patients with LGG. We tested our hypothesis using GEPIA's log-rank test and included the hazard ratio (HR) and 95% confidence intervals (CI), considering $p$-values < 0.05 statistically significant.

### TCGA and cBioPortal analysis

According to TCGA dataset, we selected 530 pathology reports for further analysis (http://www.cbioportal.org/) (*Gao et al., 2013*). Kaplan–Meier plots displayed the relationship between EMILIN/Multimerin genetic mutations, OS and DFS in patients with LGG. A log-rank test was performed to identify different survival curves. We used cBioPortal's online instructions to calculate OncoPrint, cancer type summary and gene co-expression. Additionally, a network of EMILIN/Multimerins and neighboring genes was constructed using GeneMANIA software (www.genemania.org) (*Warde-Farley et al., 2010*).

### TIMER analysis

Gene modules were used in TIMER (http://cistrome.shinyapps.io/timer) (*Li et al., 2017*) to analyze EMILIN expression levels in different cancer types. We studied the correlation between EMILIN and immune cells, including CD4+ T cells, B cells, CD8+ T cells, neutrophils, macrophages and dendritic cells.

### Metascape analysis

Metascape (http://metascape.org) (*Tripathi et al., 2015*) was used to conduct pathway and enrichment analysis of EMILIN family members and neighboring genes associated with EMILIN alterations. Gene Ontology (GO) terms (biological process, cellular component and molecular function categories) and Kyoto Encyclopedia of Genes and Genomes (KEGG) pathways were used in enrichment analysis. We only considered terms with a $p$-value < 0.01, minimum count of three and enrichment factor >1.5 to be significant (*Chen et al., 2019*). The most statistically significant term within a cluster was chosen as the representative. A select subset of enriched terms was rendered as a network plot to further determine the relationship between terms; terms with a similarity >0.3 were connected by edges. Protein–protein interaction (PPI) enrichment analysis was performed using the following databases: BioGrid, InWeb_IM and OmniPath. Additionally, the Molecular Complex Detection (MCODE) algorithm was applied to identify densely connected network components.

## RESULTS

### Expression analysis of EMILIN/Multimerin family members in patients with LGG

To explore the prognostic and potential therapeutic value of varying EMILIN/Multimerin family members in LGG patients, the gene expression was analysed by *ONCOMINE*

| Analysis Type by Cancer | Cancer vs. Normal EMILIN1 | | Cancer vs. Normal EMILIN2 | | Cancer vs. Normal EMILIN3 | | Cancer vs. Normal MMRN1 | | Cancer vs. Normal MMRN2 | |
|---|---|---|---|---|---|---|---|---|---|---|
| Bladder Cancer | 1 | 5 | | | 3 | | | | | 3 |
| Brain and CNS Cancer | 7 | | 2 | | 5 | 2 | 1 | | 1 | 5 |
| Breast Cancer | 3 | 1 | 3 | 13 | 2 | 6 | | 13 | 1 | 8 |
| Cervical Cancer | | 1 | | 2 | 1 | | | 1 | | 1 |
| Colorectal Cancer | | 2 | 2 | 7 | 8 | 1 | | 16 | | 9 |
| Esophageal Cancer | 3 | | 4 | | | 2 | 2 | 1 | 3 | 1 |
| Gastric Cancer | 3 | | 4 | 1 | 9 | 1 | | 16 | | |
| Head and Neck Cancer | 1 | 1 | 5 | | | | | 13 | 1 | 3 |
| Kidney Cancer | | 5 | 2 | | 2 | | | 1 | | 5 |
| Leukemia | 2 | 1 | 1 | 5 | | | 2 | | 1 | |
| Liver Cancer | 3 | 1 | | | 2 | | | 7 | | |
| Lung Cancer | 3 | 1 | | 2 | 2 | | | 10 | | 11 |
| Lymphoma | 9 | | 10 | 2 | | 1 | | | 3 | 1 |
| Melanoma | | | 1 | | 1 | | | 1 | | 1 |
| Myeloma | | | | | | | | | | |
| Other Cancer | 5 | 4 | 1 | 2 | 1 | | 1 | 5 | | 6 |
| Ovarian Cancer | | 3 | 1 | 1 | | | | 1 | | 1 |
| Pancreatic Cancer | 1 | | | 1 | 1 | | | 1 | | |
| Prostate Cancer | | 6 | 2 | | 1 | 5 | 1 | 1 | | 1 |
| Sarcoma | 7 | 1 | | | 2 | | | 2 | | 11 |
| Significant Unique Analyses | 48 | 31 | 39 | 35 | 40 | 17 | 8 | 89 | 10 | 66 |
| Total Unique Analyses | 413 | | 358 | | 310 | | 429 | | 359 | |

1 5 10    10 5 1
%

**Figure 1 The expression levels of EMILIN/Multimerin family members in different types of cancers (ONCOMINE).** The graph shows the numbers of datasets with statistically significant mRNA over-expression (red) or down-regulated expression (blue) of the target gene. The threshold was designed with the following parameters: $p$ value of 0.05 and fold change of all.

database. The expression of EMILIN/Multimerins in different types of cancers are shown in Fig. 1 and Table 1. In the research of *Lee Brain* (*Lee et al., 2006*), *Gutmann et al. (2002)*, *TCGA*, *Sun et al. (2006)*, *French et al. (2005)* and *Murat et al. (2008)*, we all found the significantly over-expression of *EMILIN1* in Brain and CNS Cancers in parallel to normal tissues. In the datasets of *Lee et al. (2006)* and *Bredel et al. (2005)*, we observed highly increase in *EMILIN2* expression in Brain and CNS Cancers than normal tissues. The results from *TCGA dataset* and *Beroukhim dataset* (*Beroukhim et al., 2007*) indicated that *EMILIN3* was significantly up-regulated in Brain and CNS Cancers than in normal tissues. Furthermore, the expression of *MMRN1* was shown to be considerably higher in Brain and CNS Cancers than normal tissues in the *Beroukhim dataset*

**Table 1 The transcription levels of ENILIN/Multimerin family members in between different types of brain and CNS cancer and normal tissues (ONCOMINE).**

| | Types of brain and CNS cancer vs. normal | Fold change | *t*-Test | *p*-Value | Ref. | PMID |
|---|---|---|---|---|---|---|
| EMILIN1 | Glioblastoma vs. normal | 24.769 | 18.827 | 2.04E−9 | Lee brain | 16697959 |
| | Pilocytic astrocytoma vs. normal | 3.233 | 3.030 | 0.008 | Gutmann brain | 11929829 |
| | Brain glioblastoma vs. normal | 1.529 | 11.066 | 6.72E−10 | TCGA brain | |
| | Glioblastoma vs. normal | 6.401 | 8.441 | 8.00E−11 | Sun brain | |
| | Diffuse astrocytoma vs. normal | 4.447 | 3.147 | 0.007 | Sun brain | 16616334 |
| | Anaplastic oligodendroglioma vs. normal | 1.541 | 4.581 | 4.82E−5 | French brain | 16357140 |
| | Glioblastoma vs. normal | 1.974 | 7.018 | 9.33E−5 | Murat brain | 18565887 |
| | Glioblastoma vs. normal | 1.974 | 7.018 | 9.33E−5 | Murat brain | 18565887 |
| EMILIN2 | Glioblastoma vs. normal | 2.942 | 6.257 | 1.50E−4 | Lee brain | 16697959 |
| | Glioblastoma vs. normal | 5.277 | 6.979 | 2.68E−4 | Bredel brain 2 | 16204036 |
| EMILIN3 | Malignant glioma, NOS vs. normal | 1.041 | 2.764 | 0.008 | TCGA brain 2 | |
| | Brain glioblastoma vs. normal | 1.126 | 19.817 | 1.68E−67 | TCGA brain 2 | |
| | Brain astrocytoma vs. normal | 1.047 | 4.821 | 4.80E−6 | TCGA brain 2 | |
| | Glioblastoma vs. normal | 1.108 | 2.350 | 0.023 | TCGA brain 2 | |
| | Primary glioblastoma vs. normal | 1.051 | 4.394 | 1.16E−5 | TCGA brain 2 | |
| | Anaplastic astrocytoma vs. normal | −1.027 | −1.884 | 0.046 | Beroukhim brain | 18077431 |
| | Anaplastic oligoastrocytoma vs. normal | −1.389 | −4.489 | 0.001 | Bredel brain 2 | 16204036 |
| MMRN1 | Anaplastic oligodendroglioma vs. normal | 1.089 | 2.368 | 0.027 | Beroukhim brain | 18077431 |
| MMRN2 | Glioblastoma vs. normal | 3.329 | 16.713 | 7.77E−14 | Lee brain | 16697959 |
| | Primary glioblastoma vs. normal | −1.215 | 10.058 | 1.59E−17 | Beroukhim brain | 18077431 |
| | Secondary glioblastoma vs. normal | −1.121 | −2.438 | 0.013 | Beroukhim brain | 18077431 |
| | Glioblastoma vs. normal | −1.253 | −3.730 | 0.003 | TCGA brain 2 | |
| | Brain astrocytoma vs. normal | −1.127 | −5.141 | 1.47E−6 | TCGA brain 2 | |
| | Brain glioblastoma vs. normal | −1.438 | −43.327 | 3.30E−185 | TCGA brain 2 | |

(*Beroukhim et al., 2007*). The result from *Lee dataset* (*Lee et al., 2006*) showed that *MMRN2* was over-expressed in Brain and CNS Cancers, respectively.

In contrast with normal tissues, *Beroukhim dataset* (*Beroukhim et al., 2007*) and *Bredel research* (*Bredel et al., 2005*) was founded that significantly lower expressions of *EMILIN3* in Brain and CNS Cancers. And *Beroukhim dataset* (*Beroukhim et al., 2007*) and *TCGA dataset* showed that there were lower in *MMRN2* expression in Brain and CNS Cancers compared to normal tissues.

Multiple datasets showed significantly higher expressions of EMILIN/Multimerin family members in Brain and CNS Cancers. The expression levels of EMILIN/Multimerins in the Diffuse Astrocytoma and Anaplastic Astrocytoma which were the subtypes of LGG.

## Expression levels of EMILIN/Multimerins in diffuse astrocytoma and anaplastic astrocytoma subtypes of LGG

Oncomine database analysis explored the expression of EMILIN/Multimerins in LGG subtypes. We found that EMILIN1 and EMILIN2 were overexpressed in diffuse astrocytoma compared to normal brain tissue. Additionally, there were higher expression

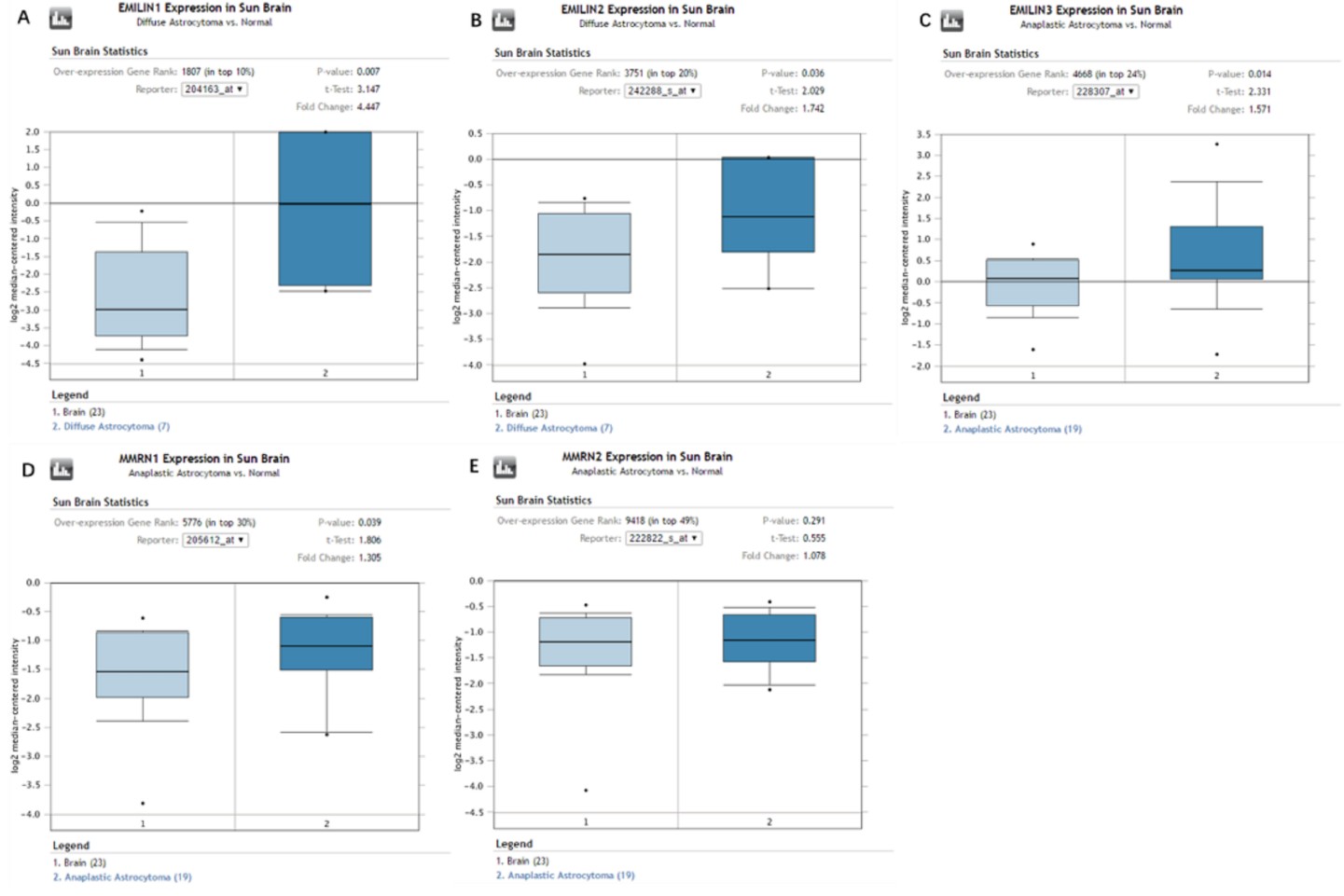

**Figure 2** **The expression of EMILIN/Multimerins in the subtypes of LGG and normal brain tissues (ONCOMINE).** (A) The expression of EMILIN1 in diffuse astrocytoma and normal brain. (B) The expression of EMILIN2 in diffuse astrocytoma and normal brain. (C) The expression of EMILIN3 in anaplastic astrocytoma and normal brain. (D) The expression of MMRN1 in anaplastic astrocytoma and normal brain. (E) The expression of MMRN2 in anaplastic astrocytoma and normal brain. The $p$ value was set up at 0.05 and fold change was defined as 2.

levels of EMILIN3 and MMRN1 in anaplastic astrocytoma compared to normal brain tissue. However, the expression of MMRN2 was not markedly higher in anaplastic astrocytoma compared to normal brain tissue (Figs. 2A–2E). Some datasets showed higher expression levels of EMILIN1, EMILIN2, EMILIN3 and MMRN1 in LGG subtypes.

## Association between the expression of EMILIN/Multimerins and tumor grades in patients with LGG

We used UALCAN to analyze the expression of EMILIN/Multimerins in LGG tumors and demonstrate the value of predicting survival rates. There was a high correlation between the expression of EMILIN/Multimerin family members and patient tumor grades (Figs. 3A–3E); patients with a more advanced tumor grade had a higher expression of EMILIN/Multimerins. These results suggest that the expression of EMILIN/Multimerin family members was significantly associated with tumor grades in patients with LGG.

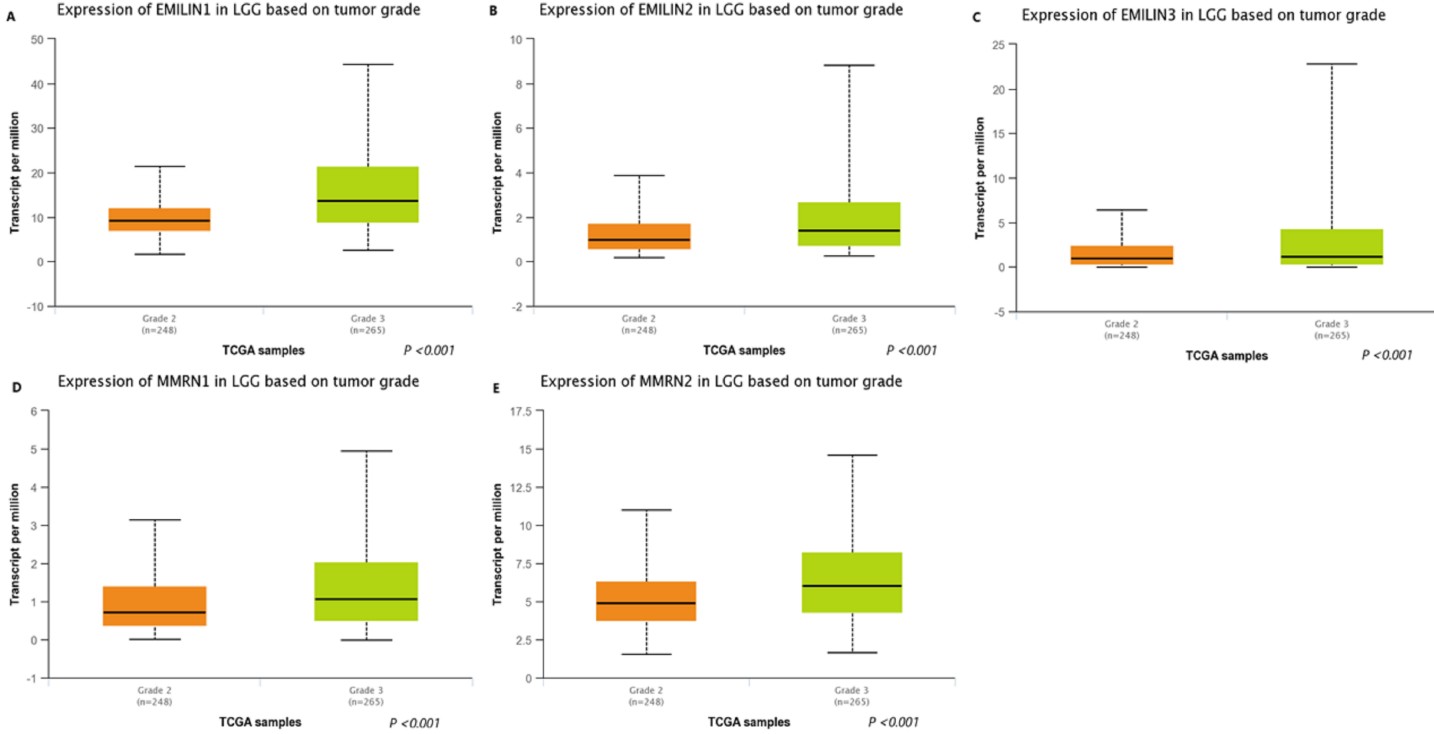

**Figure 3 The expression of EMILIN/Multimerin family members in LGG based on tumor grade (UALCAN).** (A) The expression of EMILIN1 was found to be overexpressed in grade 3 than grade 2 in LGG patients from the TCGA samples. (B) The expression of EMILIN2 was found to be overexpressed in grade 3 than grade 2 in LGG patients from the TCGA samples. (C) The expression of EMILIN3 was found to be overexpressed in grade 3 than grade 2 in LGG patients from the TCGA samples. (D) The expression of MMRN1 was found to be overexpressed in grade 3 than grade 2 in LGG patients from the TCGA samples. (E) The expression of MMRN2 was found to be overexpressed in grade 3 than grade 2 in LGG patients from the TCGA samples.

## Association between the expression of EMILIN/Multimerins and prognosis of patients with LGG

We used GEPIA to analyze the role of EMILIN/Multimerin expression levels in predicting the prognosis of LGG. The expression of most EMILIN/Multimerin family members was significantly associated with LGG patient prognosis (Figs. 4 and 5). There was a significant association between the lower expression of EMILIN/Multimerins and longer OS in patients with LGG (Figs. 4A–4E). A lower expression of EMILIN1, EMILIN2, EMILIN3 and MMRN1 was highly correlated with DFS (Figs. 5A–5E). However, MMRN2 expression levels were not correlated with DFS in patients with LGG.

Higher EMILIN/Multimerin expression levels were observed in poor LGG prognosis. We used UALCAN to demonstrate any significant associations between EMILIN/ Multimerin expression levels and a patient's LGG tumor grade (Figs. 6A–6E). This association may be used as a biomarker for predicting LGG patient survival.

## The relationship between EMILIN/Multimerin expression levels and immune infiltration levels in LGG

Immunity is closely related to the occurrence and development of tumors. TIMER analysis was applied to the relationship between EMILIN/Multimerin expression levels and

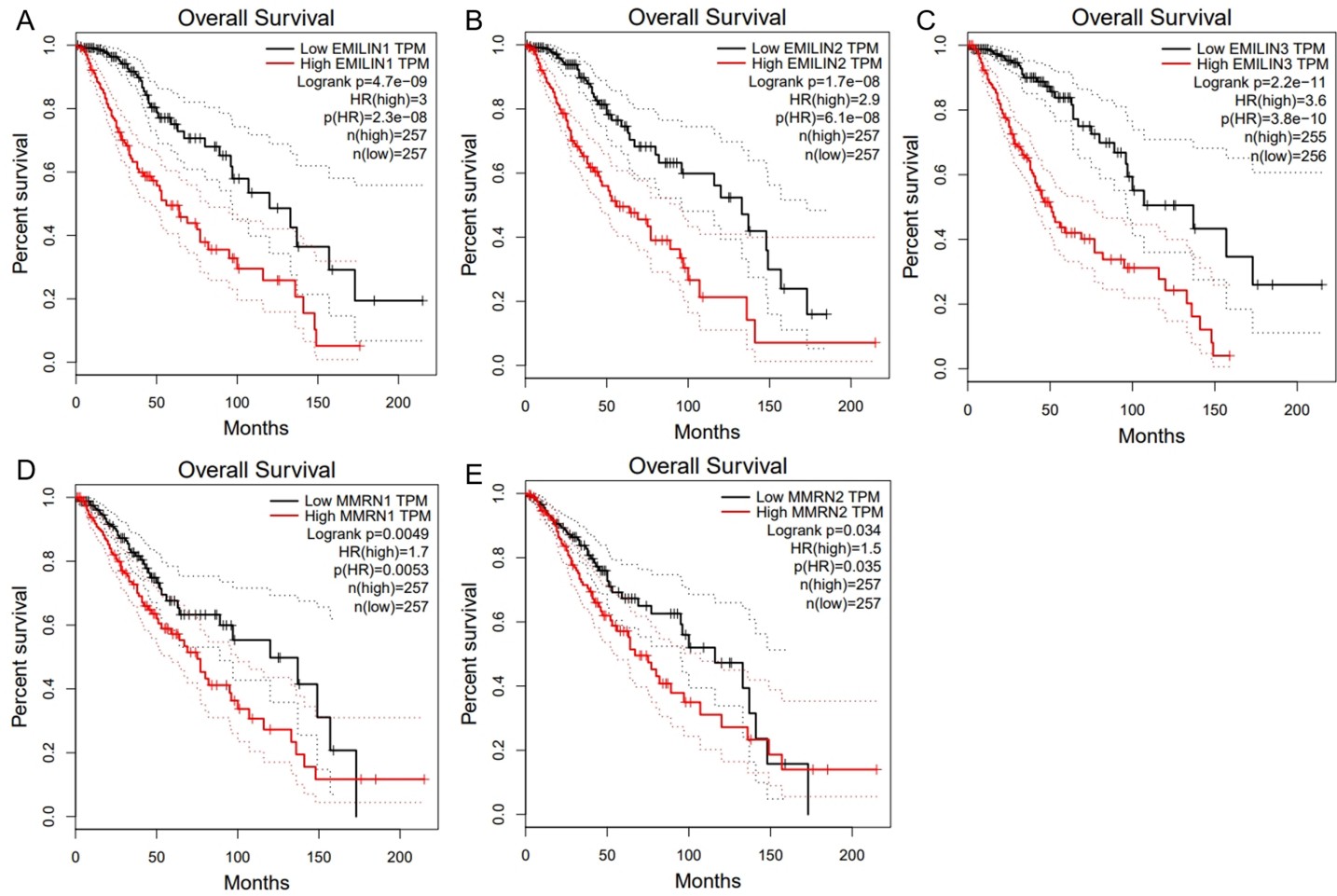

**Figure 4 The overall survival of EMILIN family members in LGG.** (A) Low expression level of EMILIN1 was associated with longer OS in LGG. (B) Low expression level of EMILIN2 was associated with longer OS in LGG. (C) Low expression level of EMILIN3 was associated with longer OS in LGG. (D) Low expression level of MMRN1 was associated with longer OS in LGG. (E) Low expression level of MMRN2 was associated with longer OS in LGG.

immune infiltration levels in LGG. EMILIN/Multimerin family members in LGG infiltrated similar immune cell profiles, showing a clear link to neutrophil, CD8+ T cell, dendritic cell, macrophage B cell and CD4+ T-cell populations (Fig. 7; Table 2). These findings suggest a correlation between EMILIN/Multimerin expression levels and immune infiltration levels in LGG.

## Exploring genetic alterations and neighboring genes of EMILIN/Multimerins in patients with LGG

The alteration frequency of EMILIN/Multimerin mutations in LGG was analyzed using cBioPortal and the results indicated five categories based on filtering (Fig. 8A). The ratios of genetic alterations in EMILIN/Multimerins range from 0.4% to 1.4% for each member (EMILIN1 0.4%, EMILIN2 1.4%, EMILIN3 0.4%, MMRN1 0.4% and MMRN2 1.4%; Fig. 8B). Additionally, we analyzed genetic alterations in EMILIN/Multimerins and their

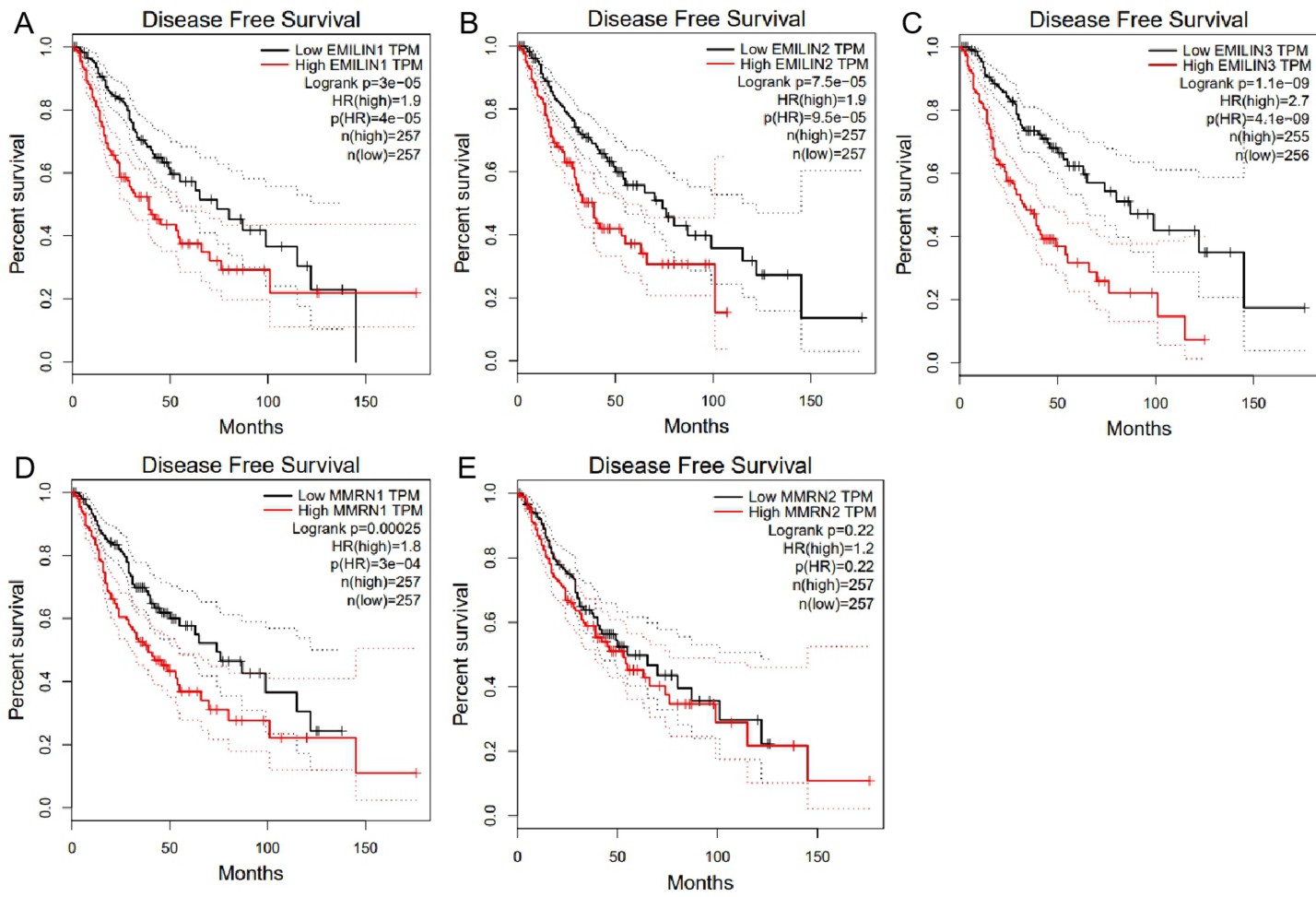

**Figure 5 The Disease Free Survival (DFS) of EMILIN/Multimerin family members in LGG.** (A) Low expression of EMILIN1 was associated with longer DFS in LGG. (B) Low expression of EMILIN2 was associated with longer DFS in LGG. (C) Low expression of EMILIN3 was associated with longer DFS in LGG. (D) Low expression of MMRN1 was associated with longer DFS in LGG. (E) The expression of MMRN2 was not significantly associated with DFS in LGG.

association with OS and DFS in patients with LGG. There was no discernable difference in OS and DFS cases with or without EMILIN/Multimerin alterations (Figs. 8C and 8D). The fifty most frequently altered neighboring genes were used to construct a network using cBioportal. These findings indicated that A2M, ACTN1, ACTN2, ALB and ALDOA were closely associated with EMILIN/Multimerin alterations and functions (Fig. 8E).

## Co-expression and interaction analyses of EMILIN/Multimerins in patients with LGG

We performed bioinformatics analysis to explore the co-expression and interaction of EMILIN/Multimerins in patients with LGG. EMILIN1, EMILIN2 and EMILIN3 were positively correlated with each other. However, there was a negative relationship between EMILIN3 and MMRN2 (Fig. 9A). Using GeneMANIA, we constructed an EMILIN/Multimerin network using the structure and function of neighboring genes. The results

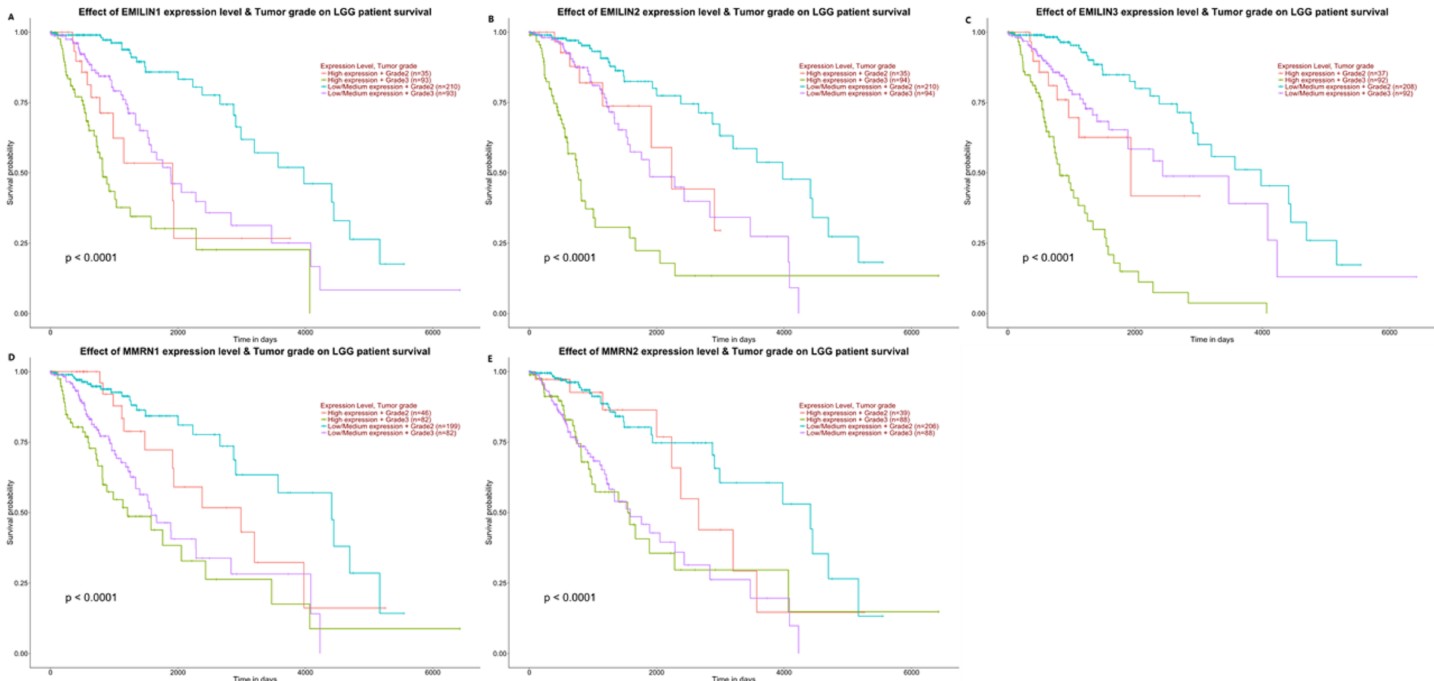

**Figure 6** The effect of EMILIN/Multimerins expression level for the tumor grade on LGG patient survival. (A) The overexpression of EMILIN1 has the poorest prognosis in higher tumor grade of LGG patients. (B) The overexpression of EMILIN2 has the poorest prognosis in higher tumor grade of LGG patients. (C) The overexpression of EMILIN3 has the poorest prognosis in higher tumor grade of LGG patients. (D) The overexpression of MMRN1 has the poorest prognosis in higher tumor grade of LGG patients. (E) The overexpression of MMRN2 has the poorest prognosis in higher tumor grade of LGG patients.

revealed twenty genes, including F5, TGFBI, EGFL7 and IL33, that were significantly associated with EMILIN/Multimerin family members (Fig. 9B).

## Functional enrichment analysis of EMILIN/Multimerins in patients with LGG

We used Metascape to analyze the functions of EMILIN/Multimerins and their neighboring genes. We performed pathway and process enrichment analysis on fifty neighboring genes (Fig. 10A; Table 3). Additionally, we constructed a network of enriched terms; a lower *p*-value was associated with terms containing more genes (Fig. 10B). We constructed a PPI network and MCODE components were identified in the gene list (Fig. 10C). We analyzed the functional enrichment of the three most significant MCODE components (Fig. 10D).

## Regulating LGG using cBioportal and co-expression analysis of EMILIN/Multimerins

Using cBioportal, we performed co-expression analysis to predict the biological functions and potential mechanisms of EMILIN/Multimerins. We selected five co-expressed genes (PHTA2, CLCL1, FSTL1, IQGAP1 and MYL12A) from the resulting analysis that were significantly associated with EMILIN/Multimerins. GEPIA's Kaplan–Meier curve was used to evaluate the function of five co-expressed genes and their impact on the survival of

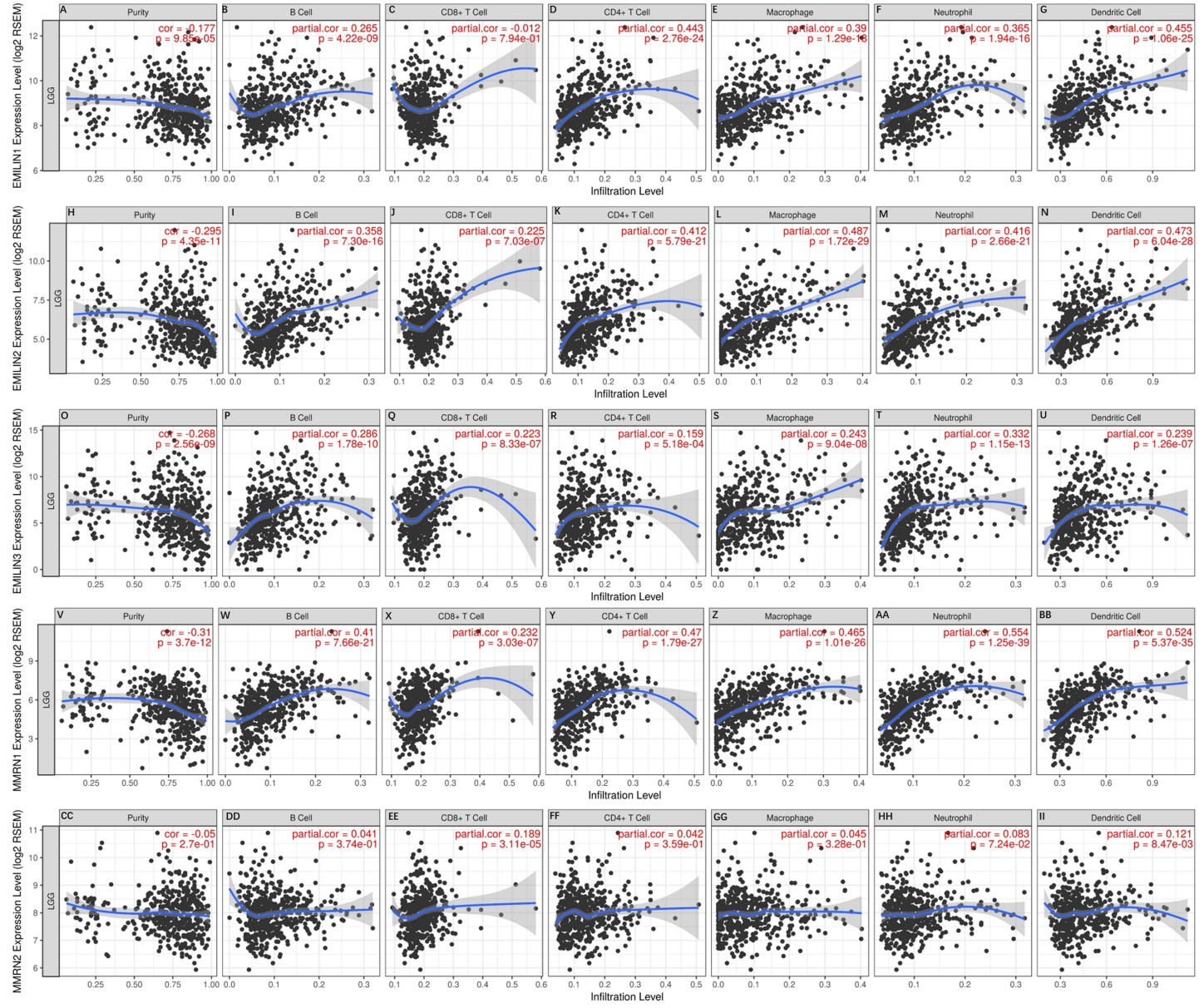

**Figure 7 Immune cell landscape of LGG compare with EMILIN/Multimerins.** (A)–(II) illustrated that scatter plots were generated using the online tool TIMER to identify different profiles of immune cells associated with investigated genes. Each dot represents a single tumor sample.

patients with LGG. Each gene was significantly related to the prognosis of LGG (Figs. 11–13).

## DISCUSSION

EMILIN/Multimerins are a unique family of glycoproteins present in the extracellular matrix (*Colombatti et al., 2012*). Previous research demonstrated the role of EMILIN/Multimerins in regulating tumor growth progression (*Andreuzzi et al., 2018*; *Rabajdova et al., 2016*) including the association of EMILIN3 in LGG survival rates (*Zeng et al., 2018*).

**Table 2 Immune cell landscape compared with gene expression.**

|  | Tumor purity | B cell | CD8+ T cell | CD4+ T cell | Macrophage | Neutrophil | Dendritic cell |
|---|---|---|---|---|---|---|---|
|  | r | r | r | r | r | r | r |
| EMILIN1 | (−) | (+) | (−) | (++) | (++) | (++) | (++) |
| EMILIN2 | (−) | (++) | (+) | (++) | (++) | (++) | (++) |
| EMILIN3 | (−) | (+) | (+) | (+) | (+) | (++) | (+) |
| MMRN1 | (--) | (++) | (+) | (++) | (++) | (+++) | (+++) |
| MMRN2 | (+) | (+) | (+) | (+) | (+) | (+) | (+) |

Note:
Categorized Pearson's product-moment correlation of immune cell landscape of LGG compared with TCGA gene expression of EMILIN1, EMILIN2, EMILIN, MMRN1, MMRN2 (TIMER). r, categorized Pearson's correlation coefficient; (--), −0.5 to −0.3, weak negative association; (−), −0.3 to 0.1, little association; (+), +0.1 to 0.3, little association; (++) +0.3 to +0.5, weak positive association; (+++), +0.5 to +1.0, strong positive association.

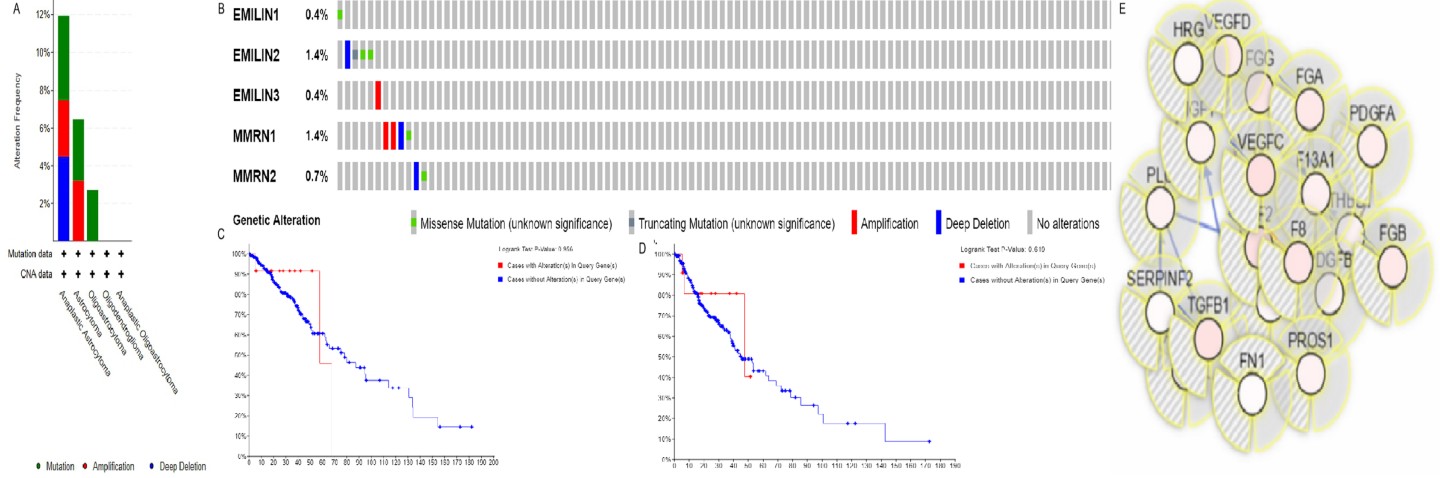

**Figure 8 Alteration frequency of EMILIN/Multimerin family members and neighbor gene network in LGG (cBioPortal).** (A) Summary of alterations in EMILIN family members. (B) OncoPrint visual summary of alteration on a query of EMILIN family members. (C) Kaplan–Meier plots comparing OS in cases with/without EMILIN family member gene alterations. (D) Kaplan–Meier plots comparing disease-free survival (DFS) in cases with/without EMILIN family member alterations. (E) Gene–gene interaction network among EMILIN family members and 50 most frequently altered neighboring genes.

LGG is a chronic disease characterized by tumor migration, infiltration of the brain's connectome and recurrence after conventional treatment (*Delgado-Lopez et al., 2017*). However, we lack knowledge regarding the relationship between EMILIN/Multimerins and LGG. In this study, we explored the expression patterns, prognostic values and potential mechanisms of EMILIN/Multimerins in LGG.

EMILIN1, the most notable member of the EMILIN/Multimerin family, was overexpressed in many organs, including blood vessels, lymphatic vessels, connective tissues, cardiovascular system and the central nervous system (*Modica et al., 2017*). The main functions of EMILIN1 include cell adhesion and migration in tumor growth (*Capuano et al., 2018*). *Modica et al. (2017)* founded that EMILIN1 can silences the RAS-ERK pathway via alpha4beta1 integrin, decreasing tumor cell growth. *Qi et al. (2019)* confirmed that EMILIN1 regulates the expression of TSPAN9, creating an anti-tumor effect in gastric cancer. However, there is no current research on the significance of

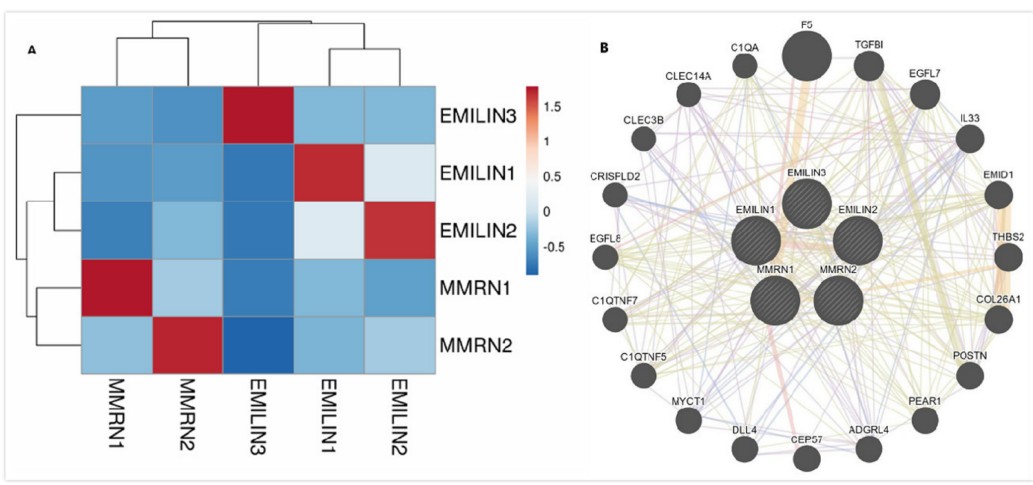

**Figure 9 Co-expression and interaction analysis of EMILIN family members at the gene and protein levels in patients with LGG (cBioPortal and GeneMANIA).** (A) Pearson correlation of EMILIN family members. (B) Protein–protein interaction network among EMILIN family members in the GeneMANIA dataset.               

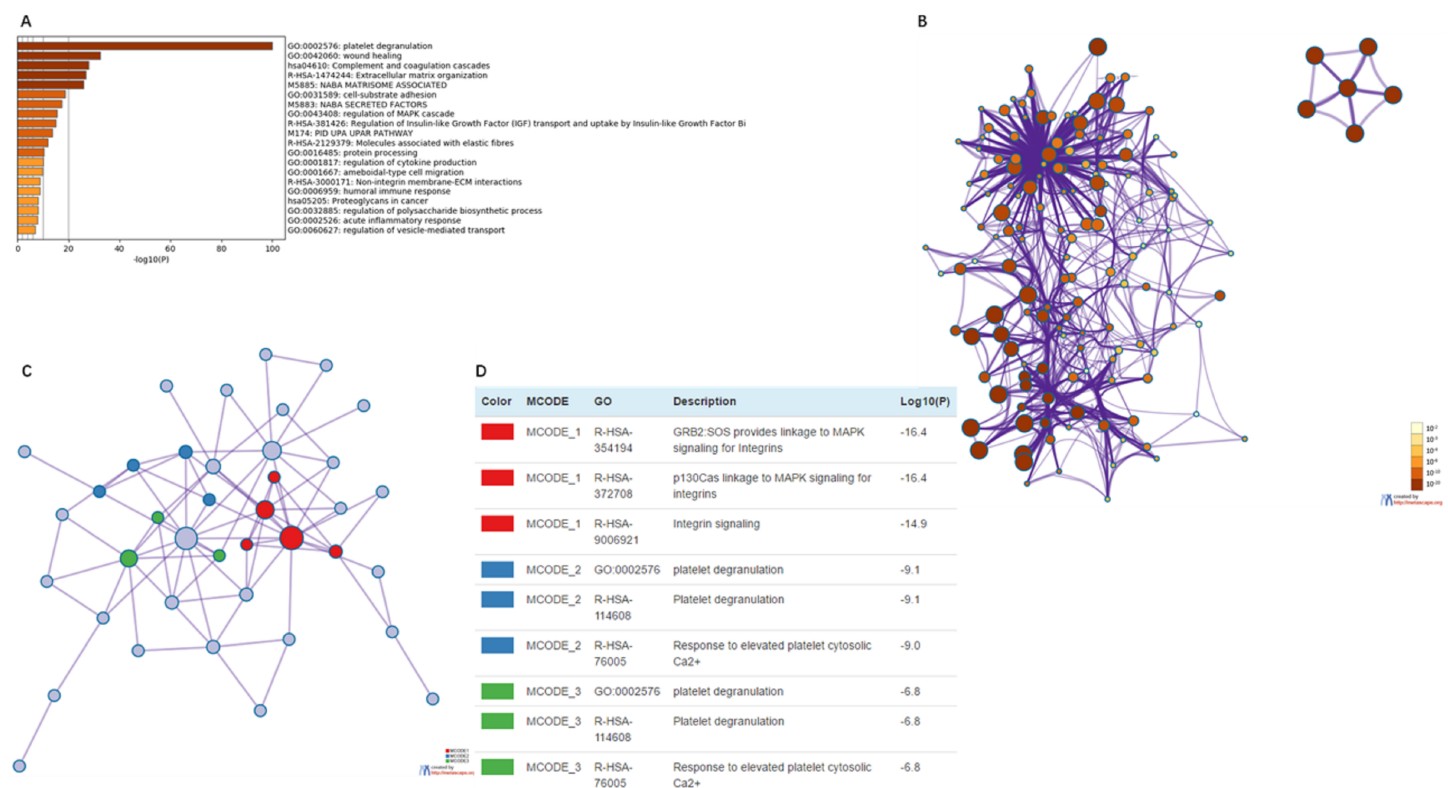

**Figure 10 The enrichment analysis of EMILIN/Multimerin family members and neighboring genes in LGG (metascape).** (A) Heatmap of enriched terms across EMILIN family members and neighboring genes, colored by *p*-values. (B) Network of enriched terms colored by *p*-value, where terms containing more genes tend to have a more significant *p*-value. (C) Protein–protein interaction (PPI) network and three most significant MCODE components form the PPI network. (D) Independent functional enrichment analysis of three MCODE components.

**Table 3 The function enrichment analysis of EMILIN/Multimerin family members and neighbor genes in LGG (metascape).**

| GO | Category | Description | Count | % | Log10 (P) | Log10 (q) |
|---|---|---|---|---|---|---|
| GO:0002576 | GO biological processes | Plateletde granulation | 45 | 91.84 | −100 | −95.98 |
| GO:0042060 | GO biological processes | Wound healing | 28 | 57.14 | −32.54 | −29.07 |
| hsa04610 | KEGG pathway | Complement and coagulation cascades | 16 | 32.65 | −27.98 | −24.62 |
| R-HSA-1474244 | Reactome gene sets | Extracellular matrix organization | 21 | 42.86 | −26.84 | −23.53 |
| M5885 | Canonical pathways | NABA MATRISOME ASSOCIATED | 26 | 53.06 | −25.87 | −22.6 |
| GO:0031589 | GO biological processes | Cell-substrate adhesion | 17 | 34.69 | −18.68 | −15.81 |
| M5883 | Canonical pathways | NABA SECRETED FACTORS | 16 | 32.65 | −17.36 | −14.54 |
| GO:0043408 | GO biological processes | Regulation of MAPK cascade | 19 | 38.78 | −15.59 | −12.88 |
| R-HSA-381426 | Reactome gene sets | Regulation of insulin-like growth factor (IGF) transport and uptake by insulin-like growth factor binding proteins (IGFBPs) | 11 | 22.45 | −14.95 | −12.3 |
| M174 | Canonical pathways | PID_UPA_UPAR_PATHWAY | 8 | 16.33 | −13.75 | −11.19 |
| RS-HSA-2129379 | Reactome gene sets | Molecules associated with elastic fibres | 7 | 14.29 | −11.96 | −9.53 |
| GO:0016485 | GO biological processes | Protein processing | 11 | 22.45 | −10.3 | −7.96 |
| GO:0001817 | GO biological processes | Regulation of cytokine production | 14 | 28.57 | −10 | −7.69 |
| GO:0001667 | GO biological processes | | 12 | 24.49 | −9.91 | −7.61 |
| R-HSA-3000171 | Reactome gene sets | | 6 | 12.24 | −8.68 | −6.5 |
| GO:0006959 | GO biological processes | | 10 | 20.41 | −8.65 | −6.49 |
| hsa05205 | KEGG pathway | Proteoglycans in cancer | 8 | 16.33 | −8.14 | −6.03 |
| GO:0032885 | GO biological processes | Regulation of polysaccharide biosynthetic process | 5 | 10.2 | −7.94 | −5.86 |
| GO:0002526 | GO biological processes | Acute inflammatory response | 8 | 16.33 | −7.82 | −5.76 |
| GO:0060627 | GO biological processes | Regulation of vesicle-mediated transport | 10 | 20.41 | −7.01 | −5.01 |

EMILIN1 in patients with LGG. In our study of EMILIN1, we observed significant links to tumor grades, overexpression in LGG and correlation between high-expression levels and poor OS and DFS.

In some solid tumors, such as gastric cancer, breast cancer, EMILIN2 can play essential functions in the tumor microenvironment, affecting tumor growth by directly binding epidermal growth factor receptor (EGFR) and lymphangiogenesis (*Marastoni et al., 2014*; *Paulitti et al., 2018*). *Paulitti et al. (2018)* confirmed that EMILIN2 have causes defective vascularization due to impaired EGFR-dependent IL-8 production affecting tumor growth. *Haage et al. (2019)* used bioinformatics analysis to identify EMILIN2 as a gene candidate distinguishing microglia from peripheral monocytes/macrophages in healthy and diseased cells. The prognostic implications of EMILIN2 in LGG are not yet understood. In this study, our results indicated that EMILIN2 had different expression levels in LGG, high-grade tumors and normal tissue. Survival curve analysis indicated that patients with LGG with a high expression of EMILIN2 were linked to poor OS and DFS.

EMILIN3 is a glycoprotein of the extracellular matrix missing the globular C1q domain (*Schiavinato et al., 2012*). *Zeng et al. (2018)* used bioinformatics analysis to study EMILIN3 as a prognostic gene in LGG, using genome-wide methylation and gene expression data.
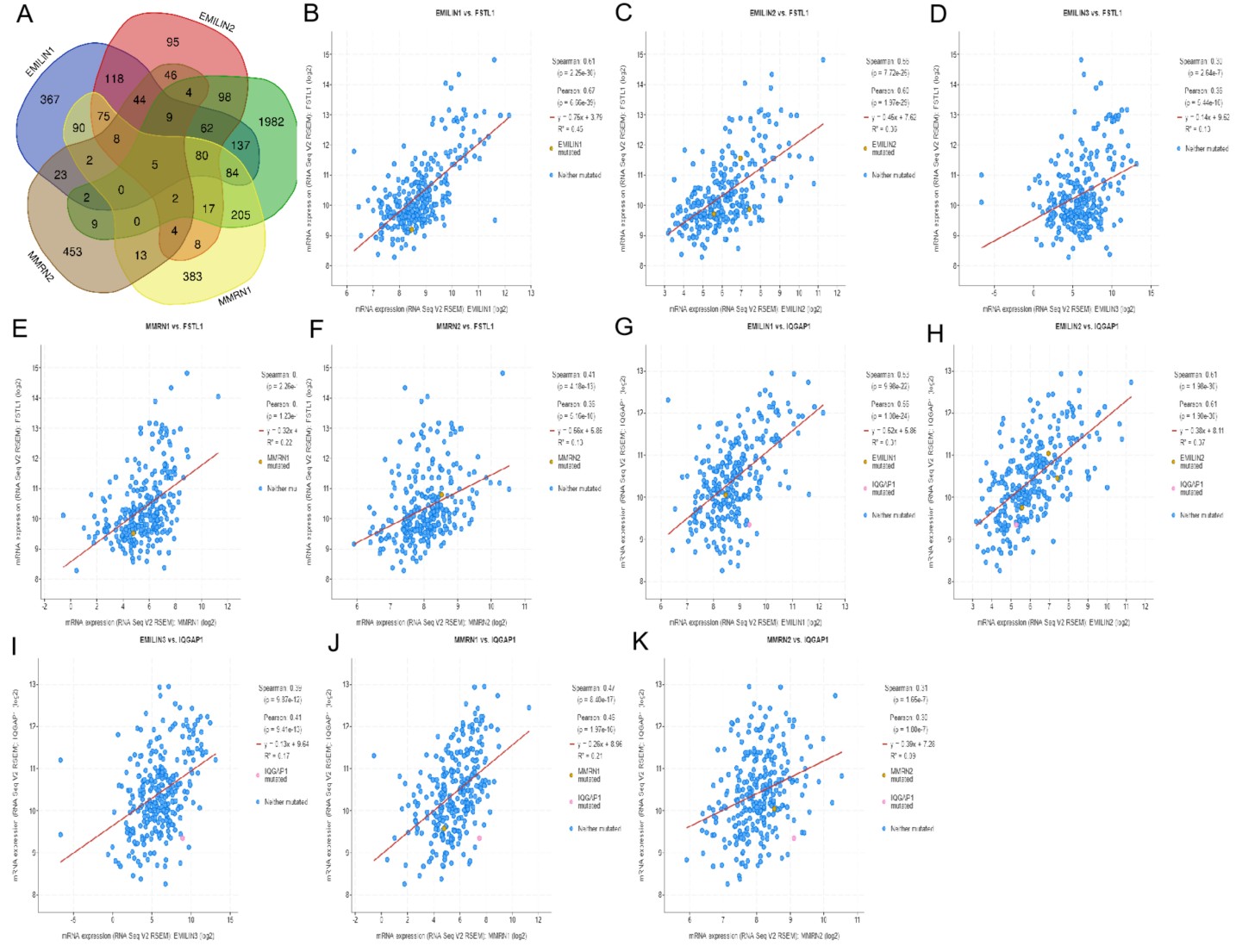

**Figure 11 The mechanism of EMILINs in regulating LGG.** (A) The Venn diagram illustrated the five co-expressed genes (PHTA2, CLCL1, FSTL1, IQGAP1 and MYL12A). (B–K) FSTL1 and IQGAP1 were significantly associated with EMILINs.

This study found a distinct EMILIN3 expression pattern in anaplastic astrocytoma; EMILIN3 was also used to identify LGG tumor grades. Additionally, elevated levels of EMILIN3 were significantly associated with shorter OS and DFS in patients with LGG.

MMRN1, a member of the EMILIN/Multimerin family, played a dynamic role in the cytoarchitectural and adhesive changes accompanying platelet aggregation and clot formation (*Laszlo et al., 2015*). *Valk et al. (2010)* found MMRN1 was involved in the development of non-small-cell lung cancer through extracellular matrix formation, apoptosis, blood vessel leakage and inflammation. The oncogenic role of MMRN1 is clear in several human cancers. In this study, the expression of MMRN1 was positively correlated with tumor grades in LGG; the overexpression of MMRN1 predicted poor prognosis in patients with LGG and indicated anaplastic astrocytoma tissue when compared to normal tissue.

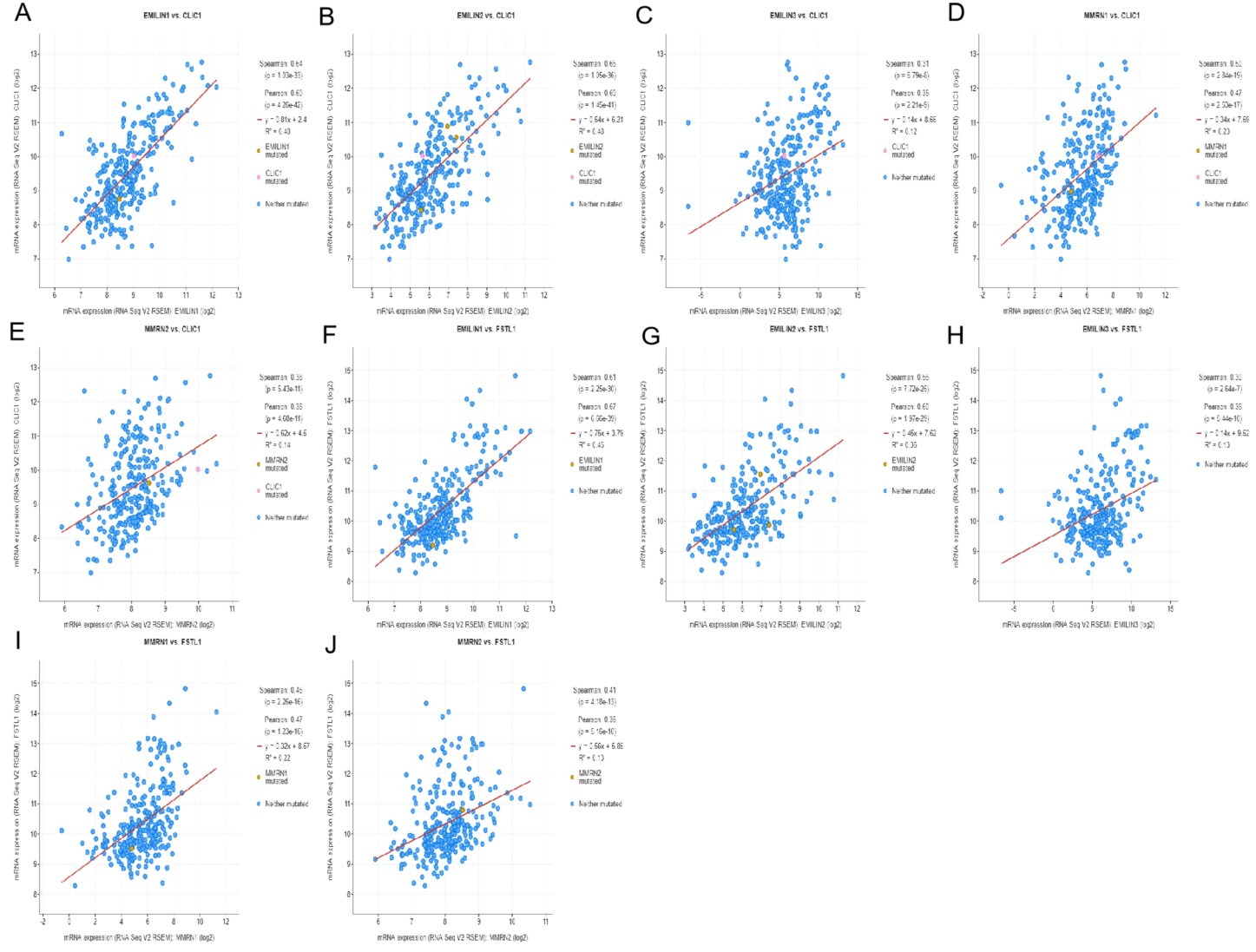

**Figure 12 The mechanism of EMILINs in regulating LGG.** (A)–(J) CLIC1 and FSTL1 were significantly associated with EMILINs.

MMRN2, an extracellular matrix molecule specifically secreted by endothelial cells, played an important role in the regulation of endothelial cell function, neo-angiogenesis and tumor progression (*Lorenzon et al., 2012*). *Noy et al. (2015)* found that MMRN2 binds to the extracellular region of CLEC14A, inhibiting sprouting angiogenesis and tumor growth. In this study, no significant difference in MMRN2 expression was observed between anaplastic astrocytoma tissue and normal tissue. However, MMRN2 expression was positively correlated with tumor grades in LGG. Overexpression of MMRN2 was related to lower OS in patients with LGG, but was not correlated to DFS.

Previous studies (*Sokratous, Polyzoidis & Ashkan, 2017*) demonstrated that immune cell activation accelerates tumor growth and progression, influencing tumor microenvironments. The key finding in this study indicated a high correlation between EMILIN/Multimerin expression and immune infiltration levels in LGG. Our results show a

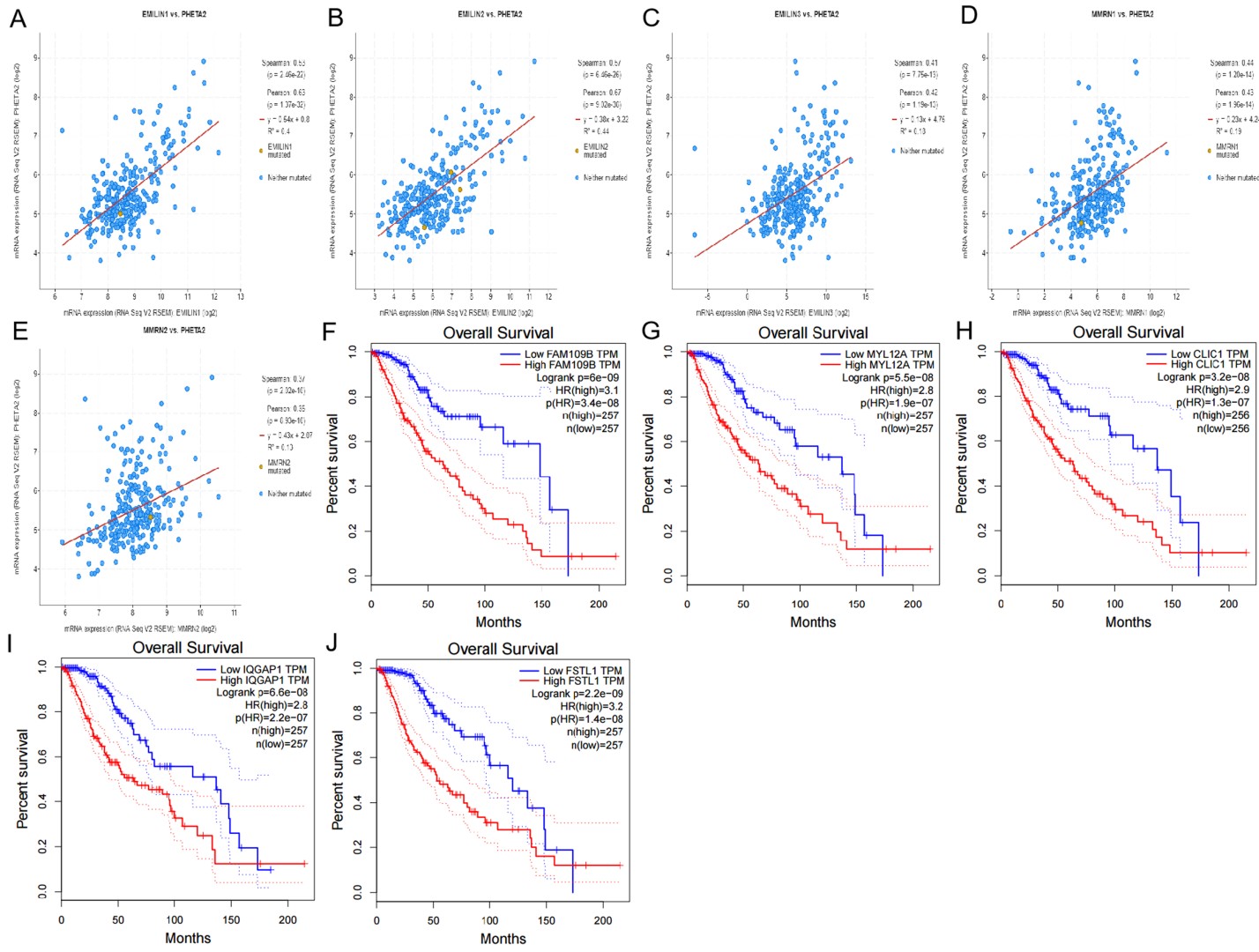

**Figure 13 The mechanism of EMILINs in regulating LGG.** (A)–(E) PHTA2 was significantly associated with EMILINs. (F)–(J) The prognostic value of five co-expressed genes in LGG patients.

positive relationship between EMILIN/Multimerin expression levels and infiltration levels of B cells, CD4+ T cells, CD8+ T cells, neutrophils, macrophages and dendritic cells. Additionally, a strong association was observed among MMRN1 and neutrophils and dendritic cells. Currently, studies illustrate that EMILIN/Multimerins display many functions regulating tumor growth and lymph node metastases. During this study, we performed co-expression and correlation analyses on EMILIN/Multimerins. We found that these family members were strongly related to each other and the constructed PPI network.

To reveal the probable mechanism of EMILIN/Multimerins in LGG, we developed an alteration frequency in anaplastic astrocytoma, astrocytoma, oligoastrocytoma, oligodendroglioma and anaplastic oligoastrocytoma. We calculated the percentage of genetic alterations in EMILIN/Multimerin family members using the TCGA dataset and

found they varied from 0.4% to 1.4%. The mutation analysis showed no significant prognostic value in OS and DFS in patients with LGG, although the underlying mechanism is still unclear. We also constructed a network of EMILIN/Multimerin family members and 50 neighboring genes. We selected five co-expressed genes: PHTA2, CLCL1, FSTL1, IQGAP1 and MYL12A. We found that these genes significantly correlated with EMILIN/Multimerins and predicted LGG patient prognosis. Due to a current lack of evidence, additional experiments are needed to investigate the role of EMILIN/Multimerins in LGG, including its potential mechanism in diagnostic and prognostic evaluation.

## CONCLUSIONS

In conclusion, our study revealed that EMILIN/Multimerins were overexpressed at the mRNA level and positively associated with LGG tumor grades. EMILIN/Multimerin expression levels were systematically analyzed to evaluate their prognostic value in LGG and find effective strategies for diagnosis and treatment. These results illustrate the potential for EMILIN/Multimerins to serve as biomarkers in patients with LGG.

### Funding
The authors received no funding for this work.

### Competing Interests
The authors declare that they have no competing interests.

### Author Contributions
- Yonghui Zhao conceived and designed the experiments, performed the experiments, analyzed the data, prepared figures and/or tables, authored or reviewed drafts of the paper, and approved the final draft.
- Xiang Zhang conceived and designed the experiments, performed the experiments, prepared figures and/or tables, and approved the final draft.
- Junchao Yao performed the experiments, analyzed the data, authored or reviewed drafts of the paper, and approved the final draft.
- Zhibin Jin analyzed the data, prepared figures and/or tables, and approved the final draft.
- Chen Liu conceived and designed the experiments, authored or reviewed drafts of the paper, and approved the final draft.

### Data Availability
    The raw measurements are available in the Supplemental Files.

### Supplemental Information
Supplemental information for this article can be found online at http://dx.doi.org/10.7717/peerj.8696#supplemental-information.

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
