# Peer review of "Expression patterns and the prognostic value of the EMILIN/Multimerin family members in low-grade glioma"

_PeerJ, doi:10.7717/peerj.8696_

## Round 0.1 · original submission · Major Revisions

All the critiques of all reviewers should be carefully addressed and the manuscript should be revised accordingly

Reviewer 1 ·

Basic reporting

This manuscript was written in technically correct English but needs a lot of proofreading such as line 58-61, line 78-79, line 117 (Figure title is not clear, language problem), line 145-149 (formatting is not consistent throughout the main test).
Some figures are low resolution: Figure 2, Figure 7.

Experimental design

The authors did not provide the essential information in the material section for the readers to understand how different criteria were set and how statistical significance was determined. Without such information, it’s hard to reproduce the finding. Examples include:
1) Line 101-109, what was the statistical method used to test the significance.
2) Line 156-162, the method section is not well described what analysis was used to explore the co-expression and interaction.
3) Line 174-176, no information about what criteria were used to select the co-expressed genes.

Validity of the findings

The authors performed a lot of bioinformatic analysis without spending enough time explaining the data in the results section which leaves the readers with confusion. Examples include:
1) line 90-91, In this section, the authors stated in some datasets, the expression levels of EMILIN3. MMRN2 are upregulated, while in some other datasets the expression of the same gene is lower. This controversy cannot lead to the conclusion that a significantly higher expression of EMILIN/Multimerin family members was found in Brain and CNS Cancers in multiple datasets.
2) Line 126-127, data shown in Figure 6 cannot demonstrate the hypotheses stated by the authors
3) Line 136-138, not enough data interpretation to support the conclusion.
4) Line 172-180, this is an overstatement. The authors did not clarify what does the data indicate and there is no connection between the data presented here and mechanism.

Reviewer 2 ·

Basic reporting

In this work, the authors used bioinformatics tools to analyze the expression levels of EMILIN/Multimerins in low-grade glioma and evaluated their potential value as prognostic biomarkers.

English language needs improvement as further elaborated in general comments.

Experimental design

The aim of the study is well-defined and the methods used are appropriate.

Validity of the findings

Overall conclusion are well stated and supported by the results shown.

Additional comments

1) On lines 77-79, authors write “...Compared to normal tissues, the results from TCGA dataset and Beroukhim dataset indicated that EMILIN3 was also significantly up-regulated in Brain and CNS Cancers.” which contradicts the statement written on subsequent lines 82-83, “In contrast with normal tissues, Beroukhim dataset and Bredel research was founded that significantly lower expressions of EMILIN3 in Brain and CNS Cancers”. Also, no values are reported for EMILIN3 in Table 1 for the Beroukhim and Bredel datasets. Authors should clarify this.
2) On lines 227-229, “…Our results show that there are significant positive relationships between EMILIN/Multimerins expression level and infiltration level of B cells, CD4+ T cells, CD8+ T cells, neutrophils, macrophages, and dendritic cells…”. Significant positive relationships seem like an overinterpretation of the results as several of the members have only weak association as seen in the results presented in Figure 7 and Table 3, with little association with CD8+ T cells in particular. Strong association is only observed for MMRN1 with neutrophil and dendritic cells.
3) On lines 241-243, “…Then we mined five co-expressed genes, which were also found that these genes significantly correlated with EMILIN/Multimerins and can predict the prognosis in LGG patients.…”. Authors should add the names of the 5 genes here. Is there any indication from previous literature of a link between these genes and LGG. If so, authors should add to that in discussion.
4) Manuscript needs significant proofreading as there are spelling (ex: “distict” on line 91) and grammatical mistakes (ex: …As it “was shown” in Figure 1… on lines 71-72; …we observed “highly” increase … on line 76) at several places in the manuscript. Overall, English language needs to be improved as well, to give a few examples:
a. It is not clear what authors want to convey on lines 138-139, ‘…The incidence of genetic alteration was strongly related to cancers since we used the online cBioPortal tool to analyze the genetic alterations of EMILIN/Multimerins in LGG.…’.
b. On lines 143-144, there appears to be an incomplete sentence, “The results of the correlation between genetic alteration and prognostic worth of these members in LGG.”
c. Lines 104-105 repeats in a slightly different wording what is already stated on lines 101-103.
5) Abbreviations used without first defining it, for ex: DFS on line 36.
6) Missing reference for statement on line 43-44, “The previous data have already highlighted LGG have high heterogeneity in terms of pathological, molecular features and prognosis.”

Reviewer 3 ·

Basic reporting

Overall, this is a very nice article, there are many studies about EMILIN/Multimerin family members, but little is known about their function in GBM. This paper firstly introduced the expression level of EMILIN/Multimerin family members in Pan-cancer. Then they described the showed the correlation of EMILIN/Multimerin family members with LGG grade and patient survival. More interesting, they investigate the interactions and “driver-passengers” relationship between EMILIN/Multimerin family members and others potential genes in detail. There are only minor comments that I would have. Some more recent research article should be discussed in this paper before publication.

Experimental design

More database should be considered.

Validity of the findings

No comment

Additional comments

Overall, this is a very nice article, there are many studies about EMILIN/Multimerin family members, but little is known about their function in GBM. This paper firstly introduced the expression level of EMILIN/Multimerin family members in Pan-cancer. Then they described the showed the correlation of EMILIN/Multimerin family members with LGG grade and patient survival. More interesting, they investigate the interactions and “driver-passengers” relationship between EMILIN/Multimerin family members and others potential genes in detail. There are only minor comments that I would have. Some more recent research article should be discussed in this paper before publication.

---

## Round 0.2 · Minor Revisions

Please address remaining issues pointed by the reviewer and amend your manuscript accordingly

Reviewer 2 ·

Basic reporting

NA

Experimental design

NA

Validity of the findings

NA

Additional comments

Authors have addressed all my concerns. However, the manuscript still lacks in professional English with many spelling and grammatical mistakes. I would recommend publishing the manuscript provided authors proofread and correct for mistakes in English. Also please update Table 1, as the correction (missing datasets for EMILIN3) that authors indicated in the "Response to Reviewers" file is not included in the Table 1 of revised manuscript.

---

## Round 0.3 · accepted · Accept

Since all the remaining issues were addressed and the manuscript was revised and edited, its amended version is acceptable now.